

# Communicating Disaster Risk? An Evaluation of the Availability and Quality of Flood Maps

Daniel Henstra[1], Andrea Minano[2], Jason Thistlethwaite[2]

[1] Department of Political Science, University of Waterloo, Waterloo, Ontario, N2L 3G1, Canada

[2] School of Environment, Enterprise and Development, University of Waterloo, Waterloo, Ontario, N2L 3G1, Canada

*Correspondence to*: Daniel Henstra (dhenstra@uwaterloo.ca)

**Abstract.** One of the key priorities for disaster risk reduction is to ensure decision-makers, stakeholders, and the public understand their exposure to disaster risk, so that they can take protective action. Flood maps are a potentially valuable tool for facilitating this understanding of flood risk, but previous research has found that they vary considerably in availability and quality. Using an evaluation framework comprising nine criteria grounded in existing scholarship, this study assessed the quality of flood maps available to the public in Canadian communities located in designated flood risk areas. It found that flood maps in most municipalities (62%) are low-quality (meeting less than 50% of the criteria) and the highest score was 78% (7 of 9 criteria met). The findings suggest that a more concerted effort to produce high-quality, publicly-accessible flood maps is required to support Canada's international commitment to disaster risk reduction. Further questions surround possible weighting of quality assessment criteria, whether and how individuals seek out flood maps, and how flood risk information could be better communicated using modern technology.

## 1 Introduction

Flooding is a major global problem that affects millions of people annually. Both the frequency and magnitude of extreme floods have grown over the past few decades (Berghuijs et al., 2017), while models project increased future flooding along rivers (Alfieri et al., 2016; Winsemius et al., 2016), in coastal zones (Vitousek et al., 2017), and in urban areas (Kundzewicz et al., 2014). Countering this threat requires a strategy of *disaster risk reduction*, meaning a concerted effort to "reduce the damage caused by natural hazards…through an ethic of prevention" (UNISDR, 2018).

This strategy of disaster risk reduction is embodied in the Sendai Framework for Disaster Risk Reduction, an international agreement endorsed in 2015 by 187 United Nations members. The Framework's first priority—understanding disaster risk—exhorts member states to "develop, periodically update and disseminate, as appropriate, location-based disaster risk information, including risk maps, to decision makers, the general public and communities at risk of exposure to disaster" (United Nations, 2015, p.15). This priority supports risk-based decision making through the transparent exchange of accessible and up-to-date risk information (United Nations, 2015, p.14).





In the context of floods, this priority suggests that stakeholders must understand the probability of flooding at their location, the likely inundation zone of a flood of a particular magnitude, possible impacts on their property and assets, and measures they can take to mitigate the risk. Flood maps—cartographic depictions of geographic areas that could be flooded—are a potentially valuable tool for facilitating this understanding of disaster risk (Dransch et al., 2010). Flood maps are used for a variety of purposes (e.g., land use decisions; emergency management) and by various users (Van Alphen et al., 2009). As a result, one kind of flood map is not suitable for all purposes, so they must be designed with consideration to who will be using them and for what purpose (Sayers et al., 2013).

One important purpose of flood maps is to communicate risk to public audiences (Hagemeier-Klose and Wagner, 2009; Kellens et al., 2009). Flood maps used for risk communication generally seek to raise public awareness about flood impacts, impart flood preparedness advice, and increase transparency about government actions for reducing flood risk. However, flood maps designed for this purpose must ensure that intended audiences are able to understand and correctly interpret the information presented (Kellens et al., 2009; Van Kerkvoorde et al., 2018).

Since one of the key principles of modern flood risk management is to enable citizens to understand and act on flood risk (Sayers et al., 2013), this study analyses the suitability of existing flood maps as tools for communicating risk to the public. To this end, the article identifies key characteristics that experts associate with flood maps suitable for public audiences and combines them into an evaluative framework. This framework is then applied to evaluate publicly and freely available web-based flood maps of Canadian communities. It finds that flood maps are often difficult to locate through online searches and most flood maps available to Canadians are not suitable for communicating flood risk to the public.

The paper begins by drawing insights from existing literature about types of flood maps and previous studies that have sought to evaluate their quality. It then sets the context for the present study by providing a short history of flood mapping in Canada. The fourth section describes the study's methods of data collection and analysis, including how the maps were located and catalogued and the assessment criteria used to evaluate their quality. Sect. 5 presents the results of the analysis, describing the availability and quality of flood maps in Canada, and this is followed by a discussion of the key findings and implications. The paper concludes with recommendations for future policy and research on flood mapping.

## 2. Literature Review

There are two main types of flood maps, which can be differentiated from one another based on their purpose and content (Canada, 2017, p.5; EU Environment, 2007, p.11). Flood *hazard* maps indicate geographic areas, typically along waterways and coasts, that could be covered by a flood of a particular magnitude (e.g., the "100-year flood" or "1% annual exceedance probability"). They are typically used to support planning and engineering functions, such as setting zoning regulations and enforcing development standards (Porter and Demeritt, 2012). In the Canadian province of Ontario, for example, flood hazard maps are created by Conservation Authorities—regional watershed management agencies empowered by provincial legislation—and are used to regulate development in flood-prone areas along waterways.



Flood hazard maps sometimes also include additional information, such as the type of flood, flood extent, water depths, and flow velocity, which can be useful for raising public awareness about flooding (Paine and Watt, 1992). As Kjellgren (2013, p.1857) argues, "by providing a visual image of the foreseen consequences of flooding, flood hazard maps can enhance people's knowledge about flood risk, making them more capable of an adequate response." Although flood hazard maps

provide a rational basis for public policies and administrative decisions, they typically contain highly technical data, lack information on potential adverse consequences associated with flooding, and fail to distinguish between different flood sources. These characteristics limit their utility for strengthening public understanding of flood risk.

Flood *risk* maps include flood hazard information, but also depict assets at risk (e.g., structures; critical infrastructure) and include indicators of the adverse consequences associated with floods, typically denoted in terms of households affected,

economic activity likely to be affected, and so on (Stevens and Hanschka, 2014, p.909). With their enhanced detail, flood risk maps are valuable for stimulating policy dialogue about flood risk management, supporting decisions about strategic investments in structural and non-structural mitigation, informing insurance underwriting, and increasing public awareness of flood risk (Büchele et al., 2006; Marco, 1994).

Good flood maps are important for a number of reasons. First, outdated or poor-quality flood maps allow for faulty planning

decisions that put people and property at risk (Keller et al., 2017; Reid, 2014). Second, communicating flood risk to stakeholders and the public in an effective way is important to build trust in the information disseminated by authorities and to motivate those at risk to take protective actions. Finally, flood maps that effectively assist people in understanding their risk are important to legitimate potentially contentious decisions around disaster risk reduction, such as relocating households out of harm's way (Kellens et al., 2009). For these reasons, evaluating the quality of flood maps is an important imperative.

Flood map analysis and evaluation has a relatively strong scholarly foundation, as illustrated by a number of studies that have assessed flood maps over the past decade. Hagemeier-Klose and Wagner (2009), for instance, compared flood maps across five European countries—Germany, Austria, Switzerland, the Netherlands, and Great Britain—in order to evaluate them in terms of readability, design, and content. They found considerable variation in the comprehensiveness and complexity of the information presented and differences in the terminology used to describe flood hazard probability.

In a more comprehensive analysis, de Moel et al. (2009) assessed the availability and content of flood maps in 29 European countries. They found that most states have flood maps covering the bulk of their territory but noted that very few have produced flood risk maps that include information on the consequences of flooding, such as economic damage or the number of people likely to be affected. These findings confirmed those of an earlier study of European flood mapping practices, which observed, "maps that illustrate possible consequences of inundations or information that helps to mitigate flood damages are

rare" (Merz et al., 2007, p.234).

In Canada, Stevens and Hanschka (2014) collected flood hazard maps from every municipality in the province of British Columbia and evaluated them based on 32 good mapping practices, such as whether they included a legend, indicated the floodway boundary, included the flood elevation for different probabilities, and showed the boundaries of individual property



parcels. They found that only 43 per cent of municipalities possessed a flood hazard map and most of these maps were of poor quality for land use decision-making, with no map containing more than 15 of the 32 assessment criteria (i.e., >47%).

## 3. Study Context

Canada is a large and geographically diverse country, with regional exposure to all forms of flooding, including riverine (fluvial) inundation, coastal flooding caused mainly by storms and storm surge, and surface water (pluvial) flooding caused by heavy precipitation, which flows into streets and affects nearby structures (Burn and Whitfield, 2016; Tucker, 2000). More than 80 significant flood disasters have affected various parts of Canada since the year 2000, and extreme rainfall alone caused more than $20 billion in losses in urban areas from 2003 to 2012 (Kovacs and Sandink, 2013; Public Safety Canada, 2015).

Canada is also a federal state, in which sovereign authority is constitutionally divided between one national government and ten provincial governments. Flood risk management (including flood mapping) is overseen predominantly by the provinces, which set regulatory standards for development, fund structural mitigation works, and provide disaster assistance to affected communities. The federal government plays an important role by, for example, providing forecasts of weather conditions that could lead to flooding, monitoring flood hazards from the Government Operations Centre, funding small flood mitigation projects, and contributing to post-flood disaster assistance. Finally, local governments serve several key flood risk management functions, including using zoning by-laws to direct development away from flood-prone areas, issuing flood warnings when conditions seem imminent, and subsidizing property-level flood protection measures such as backflow preventers.

The most concerted flood mapping effort in Canada occurred under the Flood Damage Reduction Program (FDRP), an intergovernmental initiative that operated between 1975 and 1999, which aimed to identify high-risk flood areas (de Loë, 2000). Through a general agreement between the Government of Canada and the provinces, the cost of flood mapping was cost-shared on a 50-50 basis, and all provinces and territories except Prince Edward Island and Yukon participated (Bruce, 1976; Watt, 1995). Nunavut was also not a part of the FDRP since it did not become an independent territory until 1999 (ECCC, 2013; INAC, 2014). Although some provinces adopted more stringent standards, the 100-year flood was used as the minimum criterion for the FDRP, which resulted in the identification of 957 "designated flood risk areas", meaning those lands that are subject to recurrent and severe flooding (ECCC, 2013).

The FDRP differentiated between large scale "engineering maps", which contained topographic contour lines and delineated floodplains for planning purposes, and small scale "public information maps", which also contained local features such as roads and buildings, as well as the extent of historic flood events. The intergovernmental agreement specified that provincial governments would direct municipalities to regulate or prohibit development in designated areas, and both the federal and provincial governments would refuse disaster assistance to these areas once the public had been made aware of the hazard (de Loë and Wojtanowski, 2001; Page, 1980).

Although the FDRP made flood risk more transparent, weak enforcement of floodplain regulations and a general unwillingness among elected politicians to refuse disaster assistance to designated areas prompted the Government of Canada to withdraw



from the initiative in 1999 (Kumar et al., 2001; de Loë, 2000). There has since been no similar intergovernmental effort to identify high-risk lands and update flood maps, nor is there a national repository of flood maps in Canada.

In recent years, however, public officials have shown renewed interest in updating existing flood maps and producing new maps to support disaster risk reduction. In 2015, for instance, the Government of Canada launched the National Disaster

Mitigation Program, a five-year, $200 million initiative to (1) focus investments on significant, recurring flood risk and costs and (2) facilitate private residential insurance for overland flooding (Public Safety Canada, 2017). One of four funding streams pertains to flood mapping, which permits provinces and territories to apply for support to develop or modernize flood maps. Flood mapping funding has been allocated in British Columbia, New Brunswick, and Ontario (Public Safety Canada, 2018a, 2018b, 2018c).

Moreover, in 2017 a Flood Mapping Committee comprising six federal departments, which was advised by a working group that included representatives from provincial governments, industry, and academia, released a Federal Floodplain Mapping Framework. Its core objective is to "facilitate a common national best practice and increase the sharing and use of flood hazard information", in order to generate a "comprehensive understanding of hazard exposure in order to inform mitigation and preventative measures" (Canada, 2017). Finally, a national roundtable hosted by the federal Minister of Public Safety in

November 2017 brought together representatives from all levels of government, Indigenous leaders, insurers, non-government organizations, and academics to launch a formal dialogue on flood risk, which identified current and accurate flood maps as a central priority (Boyer, 2017).

The starting point for any such effort, however, is a fulsome understanding of the strengths and limitations of existing flood maps in order to identify opportunities for improvement. A preliminary scan commissioned by the Government of Canada

found that most flood maps in Canada are dated—with a median age of 18 years—and that their availability is grossly uneven across the ten provinces (MMM Group Limited, 2014). This article seeks to extend this analysis by evaluating the quality of publicly available flood maps in Canada using internationally recognized principles of good practice. The focus here is on flood maps that are freely and publicly accessible online, given that "the dissemination of flood maps via the Internet is a very important way of bringing flood information to the public" especially as more people become accustomed to digital

technologies (Hagemeier-Klose and Wagner, 2009, p.572). The next section outlines the methods and analytical framework used to undertake the quality evaluation.

**4 Methods**

This study involved four general phases. First, researchers gathered a list of communities across Canada that are at high risk of flooding. Second, a thorough online search was conducted to find maps that depict flood hazards or flood risk in the selected

communities. As discussed in detail below, researchers scanned provincial, regional, and municipal government websites, followed by a search using an online search engine. Third, a comprehensive review of scholarly and grey literature was conducted to identify nine binary criteria by which the quality of publicly-accessible flood maps could be evaluated. Finally,





two researchers assessed each map using the selected criteria. To generate a summary of findings, statistics were computed based on the highest-ranking flood maps available for each community and its respective municipality.

## 4.1 Community selection

The communities selected for this study (Table 1) included the 957 designated flood risk areas compiled during the FDRP
5  (ECCC, 2013). The FDRP communities were suitable for the purposes of this research because (1) they are known to be at high risk of flooding; (2) flood maps are more likely to be available to the public in these communities; and (3) actions are likely to have been taken by governments to reduce flood risk in these communities (e.g., development regulations). Communities in Prince Edward Island, Yukon, and Nunavut are not a part of the study area since these provinces and territories did not participate in the FDRP.

| Province/Territory | Number of FDRP communities |
|---|---|
| Alberta | 16 |
| British Columbia | 216 |
| Manitoba | 25 |
| New Brunswick | 84 |
| Newfoundland and Labrador | 24 |
| Northwest Territories | 10 |
| Nova Scotia | 24 |
| Ontario | 273 |
| Quebec | 265 |
| Saskatchewan | 20 |
| **Total** | **957** |

Table 1. Number of FDRP communities by province or territory

In order to create a dataset of manageable size, but maintain the validity of the study's results, a random sample of 369
15  communities was drawn from the complete list of 957 communities (95% confidence interval; 4% margin of error). This list captured communities in all provinces and territories that were part of the FDRP (Table 2). The final list of communities was recorded in a database, with their corresponding designated flood risk area, province, present-day municipality name (i.e., 2016 Census Subdivisions as defined by Statistics Canada), and regional watershed authority (in Ontario only).

20



| Province/Territory | Number of FDRP communities |
|---|---|
| Alberta | 4 |
| British Columbia | 86 |
| Manitoba | 10 |
| New Brunswick | 35 |
| Newfoundland and Labrador | 11 |
| Northwest Territories | 5 |
| Nova Scotia | 9 |
| Ontario | 105 |
| Quebec | 98 |
| Saskatchewan | 6 |
| **Total** | **369** |

**Table 2. Number of FDRP communities used for study by province or territory**

Because some communities have changed their names and/or been amalgamated with other communities since the time of the FDRP, we matched the FDRP-designated communities with their present-day municipal boundaries. As such, though the study area covers 369 FDRP-designated areas, these are situated within 280 municipalities.

**4.2 Online search**

10   Since they vary significantly in scope and content, a "flood map" was defined operationally as one that:

- cartographically depicts flood-prone areas for at least part of the community of interest;
- labels flood-prone areas using terminology understandable to a lay audience (e.g., floodplain, flood zone, flood hazard); and,
- is published online as either a static image (e.g., PDF document; scanned image; figure in a report) or dynamic
15   interface (e.g., interactive web map).

Researchers then sought out online flood maps for each of the 280 municipalities included in the study area. Individual online searches were conducted for each municipality because there is no national repository of flood maps in Canada, and there has been no coordinated effort among regional and local governments to produce flood maps since the FDRP. For each community, one researcher looked for flood maps by:



- using an online search engine to find the community's respective provincial, municipal, and regional watershed authority website (as applicable);
- searching these government websites using terms such as "flood", "maps", "land use", "development plan";
- seeking out missing maps through relevant departmental webpages, such as Public Works, Emergency Response, Planning and Development, By-Laws, Geographic Information Systems (GIS), and Maps;
- conducting an additional Google search using both the municipality name and "flood map" (e.g., "Calgary flood map").

As a quality check, a second researcher then reviewed the first researcher's search and followed the same steps again. Once the map was found by either researcher, its hyperlink was recorded in the database for its corresponding community. If a flood map was not found by following the above steps, the map was recorded as inaccessible to a lay audience. Some flood maps depicted flood-prone areas in more than one community; in those cases, the same hyperlink was recorded in the database under multiple communities. Conversely, some communities had multiple flood maps in cases where, for example, both the provincial and municipal governments had produced and published a flood map. In these cases, both maps were stored in the database and assessed using the quality evaluation framework, but the results from only the highest-quality map were retained for analysis and discussion. The search for flood maps concluded on July 25, 2018 (hyperlinks active to this date).

### 4.3 Quality evaluation framework

Although there are several studies that assess the design of flood maps, such as the most appropriate colour scheme for depicting floodplains (Seipel and Lim, 2017), this study focused on key characteristics that international scholars highlight as important for public risk communication. In this context, the purpose of the map is to inform the public about flood risks and motivate individuals to take precautionary actions (Hagemeier-Klose and Wagner, 2009). Although some of the maps assessed for this study may not have been created with this explicit objective, our interest was to assess how well the maps that are currently available to Canadians living in flood-prone areas inform them about flood risk.

After conducting a scan of international literature, nine criteria were identified for the quality assessment framework, including: personalized experience, local context, historical context, legend legibility, flood zone legibility, explanation of technical terms, risk reduction advice, transparency about flood modelling limitations, and depiction of multiple flood hazards (Table 3). In combination, these characteristics make a flood map more effective for risk communication, because they identify an individual property's flood risk, create relatable depictions of flood impacts on communities (e.g., using photographs), assist users in understanding the flood map and its limitations, and establish connections between hazards and risk mitigation actions.



| Criterion | Description | Sources |
| --- | --- | --- |
| Personalized experience | enables users to find information specific to their property (e.g., postal code search to locate property in relation to flood hazard) | Kellens et al., 2009 |
| Local context | contains identifiable places or landmarks (e.g., major and minor roads; public buildings; neighborhood names) that help an individual visualize the likely spatial extent of flooding | Van Kerkvoorde et al., 2018 |
| Historical context | depictions of past flood events (e.g., photographs; victim testimonials) to help users understand potential impacts | de Moel et al., 2009; Luke et al., 2018 |
| Legend legibility | clear explanation of lines, symbols, colours and terminology | EXCIMAP, 2007; Fuchs et al., 2009 |
| Flood zone legibility | easy for the user to distinguish the extents of the flood hazard zone | Hagemeier-Klose and Wagner, 2009; Kellens et al., 2009; van Alphen et al., 2009; Fuchs et al., 2009 |
| Explanation of technical terms | meaning of terms is understandable to a lay audience (e.g., properties in 20-year flood zone more likely to be flooded that those in 100-year) | Hagemeier-Klose and Wagner, 2009; Meyer et al., 2012 |
| Risk reduction advice | paired with information about the consequences of flooding and preventative or precautionary actions that residents can take (e.g., install a backwater valve; buy flood insurance) | Merz et al., 2007; Meyer et al., 2012; van Alphen et al., 2009 |
| Transparency about limitations and uncertainty | provides information about types of flooding depicted and/or potential exposure of areas adjacent to the flood lines | Merz et al., 2007 |
| Depiction of multiple flood hazards | depicts all forms of flooding to which a property is exposed (e.g., coastal, riverine, and pluvial) | EA, 2010 |

**Table 3. Map assessment criteria and examples**



### 4.4 Quality assessment

The completed database included 369 FDRP-designated communities and their associated flood maps. Two researchers then divided these maps and evaluated them using the evaluation framework discussed above. The flood map was assigned a score of '1' for each criterion it met, and a score of '0' for each criterion that was not met. For example, a map that used a distinct colour to distinguish the flood-prone area from other areas received a '1' for the 'flood zone legibility' criterion. If the map was part of a report or was linked directly to additional resources (e.g., an interactive web map with a 'Help' link), then the associated resources were also considered in the quality evaluation. Once the two researchers had completed their individual assessments, they exchanged results and conducted random 'spot checks' to ensure quality control. As noted above, where multiple maps were found for a community, only the map with the highest overall quality was selected to represent the overall quality of flood maps for that community.

### 5 Results

The results described in this section focus on those communities that had available flood maps. The evaluation results for the 369 FDRP-designated communities are grouped by their respective present-day municipalities (n=280).

### 5.1 Availability of flood maps

After completing the online search, researchers found at least one flood map for 239 of 280 municipalities (85%) in the study area (Fig. 1). Flood maps were not found for 41 municipalities (15%) in the study area. Most municipalities lacking available flood maps were in Ontario. In many of those communities, researchers found maps that depicted "development regulated areas", but these did not meet the operational definition of "flood map" outlined above (i.e., map did not clearly label flood-prone areas using terminology understandable to a lay audience).



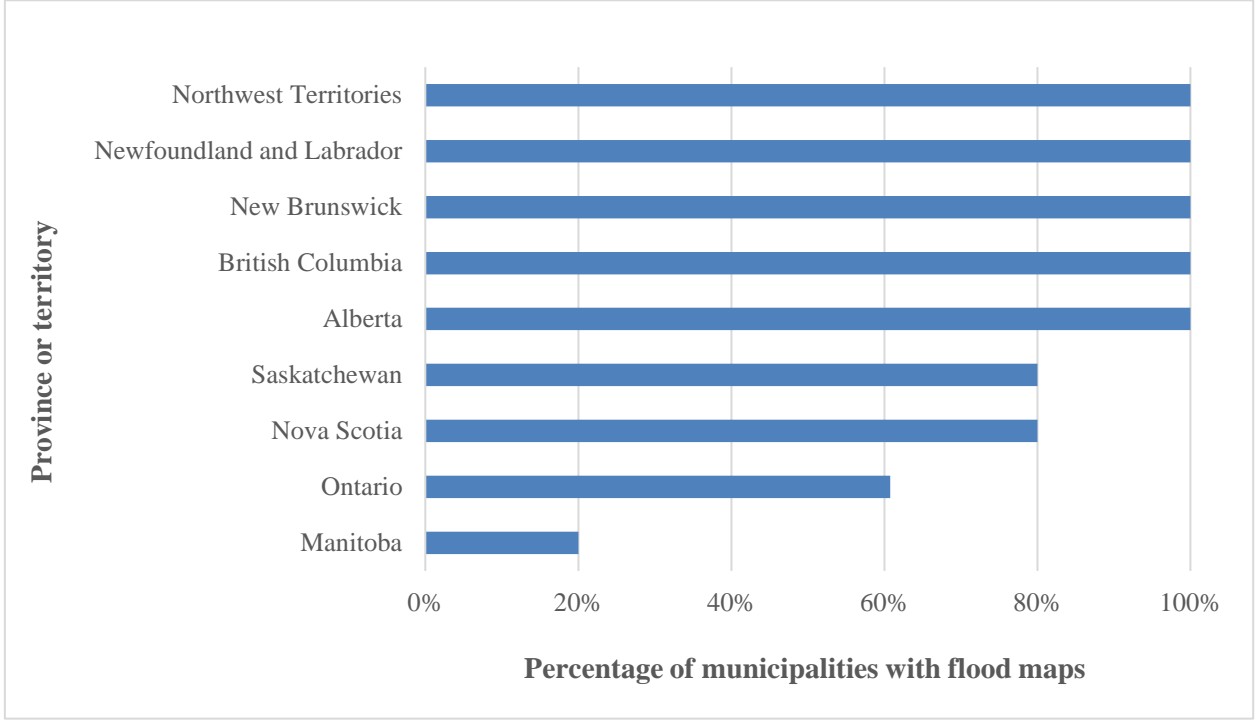

**Figure 1: Municipalities in study area with available flood maps**

## 5.2 Quality of flood maps

5   The evaluation found that flood maps in most municipalities (62%) are low quality—meeting less than 50% of the criteria (i.e. score of <4 out of 9)—and are therefore ill-suited for communicating flood risk to public audiences. Only 16% of municipalities had access to a flood map that met or exceeded five of the nine quality criteria. There were no flood maps that met all of the evaluation criteria; the highest score was 78% (7 out of 9).

Fig. 2 illustrates how different quality scores were assessed. For example, it is possible to see the weaknesses of the flood map

10   for Barriere, British Columbia, particularly in how it is difficult to distinguish the extent of the flood-prone area. The flood maps for Saint Césaire, Quebec and Pleasant Valley, Newfoundland and Labrador show the flood hazards, but include neither measures of the consequences of flooding nor information on how residents could reduce their flood risk.




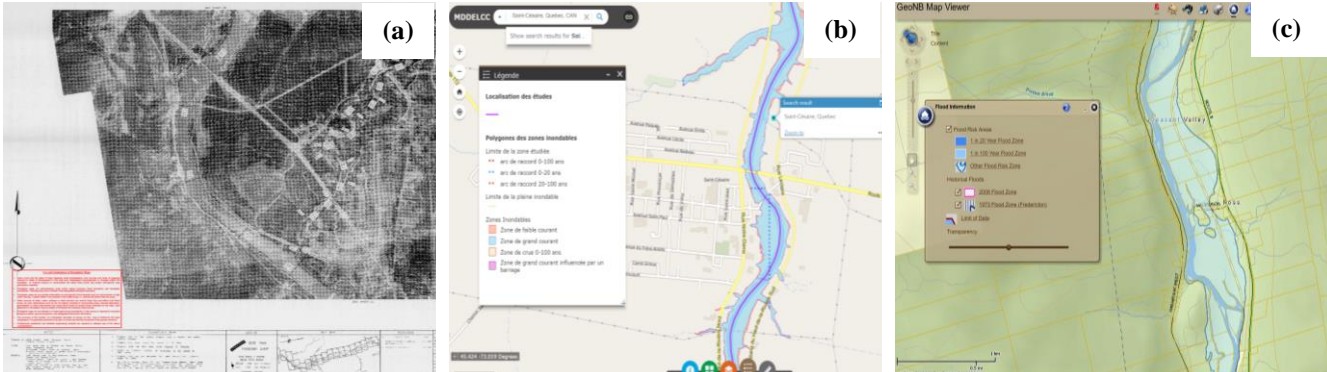

**Figure 2: Comparison of flood maps and associated quality scores. (a) Barriere, British Columbia, score of 22% (2/9) for including a legend and limitations of flood modelling; (b) Saint Césaire, Quebec, score of 44% (4/9) for including an address search, local context, legend, and legible flood zone; (c) Pleasant Valley, New Brunswick, score of 67% (6/9) for including an address search, local context, legend, legible flood zone, explanation of technical terminology, and transparency about modelling limitations.**

The municipalities with the highest-ranking flood maps were situated primarily in the provinces of Newfoundland and Labrador and New Brunswick. Newfoundland and Labrador's flood maps were created during the FDRP and ranked particularly high because the depictions of flood hazard were paired with additional locally-relevant information, such as photographs of historical floods, the number of homes evacuated, the costs of flood damage, and explanations of technical terms (e.g., 1-in-20-year flood). This confirms that the evaluation criteria are not biased towards newer maps or those generated with modern technologies (e.g., web GIS portal), but rather privilege maps that include information to improve public understanding of flood risk.

**5.3 Characteristics of flood maps**

Individual characteristics of flood maps were analysed to identify which criteria were the most frequent (Fig. 3). In summary, residents of flood-prone municipalities typically have access to flood maps that include (1) a legible legend, (2) a legible flood zone, and (3) local contextual features, such as road names and neighbourhood names.





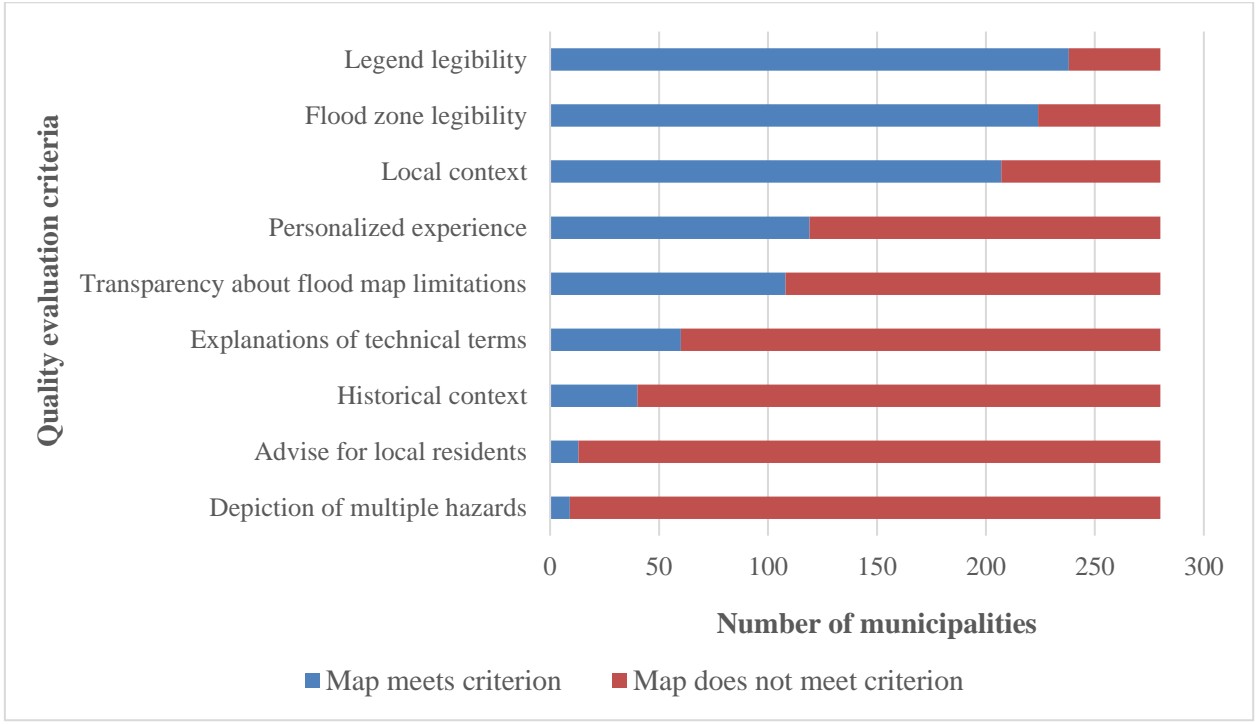

**Figure 3: Characteristics of flood maps available to municipalities**

5   By contrast, less than 45% of municipalities have access to a flood map that would enable residents to identify their individual property relative to the flood hazard. For example, despite ranking high overall, several flood maps available to Newfoundland and Labrador communities depicted general flood risk for the community, rather than identifying individual property parcels (Fig. 2). Similarly, only a small proportion of available flood maps (21%) were paired with information that would assist a user in understanding the technical terminology such as "100-year flood" and "floodplain".

10  Few flood maps (14%) included information that would engender an emotional response to flood risks among residents, such as photographs of past floods or testimonials from past flood victims, which would make them more likely to understand their risk. Even fewer (5%) connected the flood hazards portrayed on the map to actions residents could take to reduce flood impacts. Finally, only nine municipalities (3%) had access to a map that depicted multiple flood hazards. All of these maps were published by Ontario's Conservation Authorities and they depicted both coastal (lake) flood hazards and riverine floodplains.

15  Most maps found for the 280 municipalities depicted only riverine floodplains, whereas none of the maps we evaluated included information on risks from stormwater.



## 6 Discussion

The availability of flood maps in high-risk Canadian communities is poor. Unlike other states such as the United Kingdom, Canada lacks a central portal through which the public can access flood map information. Instead, flood maps are located on many different government websites, and there is inconsistency from one province to another. For example, in Alberta, New Brunswick, and Quebec, flood maps for many communities could be found in one central location, such as a provincial web GIS portal, but even in these cases, maps for individual communities and properties were difficult to find.

Riverine flood maps in British Columbia are posted on a provincial government website that groups flood maps according to regions (e.g., Vancouver Island) and designated floodplains (e.g., Cowichan River; Nanaimo River). Links on this base map then direct users to a map series index where individual flood maps can be accessed. To find a flood map for a specific community, however, users must identify their respective designated area and then look through a map series index to find their community, an effort that few individuals would be likely to expend.

Broader information access barriers experienced by the researchers included (1) difficulties finding flood extents in web GIS applications (i.e., layer turned off by default or only visible when zoomed in); (2) long load and refresh times for web applications; and (3) web applications developed with outdated software (e.g., Silverlight) that cannot be processed by modern web browsers.

The quality of flood maps in high-risk Canadian communities is also poor. Scholarly literature distinguishes between flood maps created for "experts" and for "lay users/non-experts" (EXCIMAP, 2007; Van Kerkvoorde et al., 2018). With prior knowledge of hydrology and engineering, for instance, expert users of a flood map can "handle more complex tools" and typically demand more detailed information (Van Kerkvoorde et al., 2018, p.62). By contrast, lay users require a simpler, more intuitive flood map. Most flood maps available to the public in Canada's most flood-prone communities are more suited for expert audiences than for lay users. For instance, publicly-accessible flood maps typically depict riverine floodplains and use technical terms (e.g., return periods) without providing explanations. Although these maps might be useful for experts who would understand their limitations, they are not suitable for enabling citizens to understand their disaster risk.

Historically, flood maps meant for public use were created to raise awareness about floods and encourage acceptance of government initiatives, such as the establishment of regulated floodplains (Handmer, 1980). Today, by contrast, governments are increasingly interested in shifting some responsibility for flood risk management to the public by, for example, encouraging their uptake of private flood insurance and installation of private flood protection measures (Thistlethwaite et al., 2018). In most of the municipalities we analysed, however, maps did not depict property-level flood risk or offer advice as to how property owners could reduce their risk. This highlights a strategic policy imperative for Canada: to create maps that are explicitly designed to foster public understanding of flood risks. Such maps must be compelling (i.e., property-specific; set in historical context), understandable (i.e., contextually appropriate; legible; transparent), and actionable (i.e., paired with risk reduction advice).



# 7 Conclusions

One of the key principles of disaster risk reduction, as articulated through the Sendai Framework, is that stakeholders and the public must first understand their disaster risk, as a prerequisite to reducing the damage caused by natural hazards. With this in mind, our study sought to assess the suitability of existing flood maps as tools for communicating risk to the public.

Specifically, we assessed the availability and quality of publicly-accessible, online flood maps in Canada's most flood-prone communities, namely those designated under the Flood Damage Reduction Program as being subject to recurrent and severe flooding.

Among the 280 municipalities targeted for analysis (which represented 369 FDRP-designated areas), 85% had flood maps available online. However, many of these maps were difficult to locate and most were found to be generally unsuitable for risk

communication purposes. Most lacked characteristics that would assist users in understanding information depicted on the map, such as explanations of technical terms and modelling limitations, and few included features that might motivate protective responses, such as photographs of historical floods, potential consequences for property owners, and multiple types of flooding.

As governments look to increase public involvement in flood risk management, greater effort will be required to effectively

communicate flood risk to the public. Flood maps play a significant role in this effort, but questions remain about what these maps should contain and how they should be made accessible. As evidenced in states such as Austria, Belgium, Germany, France, and the United Kingdom, surveys or focus groups could be useful tools for determining user preferences concerning map design options, such as colours, scale, terminology, and so on (Meyer et al., 2012; Van Kerkvoorde et al., 2018). A further consideration is who should produce and maintain flood maps to ensure their completeness, quality, and currency. A

decentralized approach, as in Canada, can result in considerable variation in the quality of flood maps from one community to another, and this raises questions about equity when national or regional flood risk management policies expect more responsibility to be shouldered by individuals.

The evaluation framework used in this study represents a first attempt to assess the quality of flood maps in Canada in order to identify their key strengths and limitations. Future research might involve weighting the various quality criteria based on

feedback from lay users. Moreover, in light of the barriers faced by the research team in accessing flood maps, it would be valuable to survey individuals to explore whether and how they seek out flood maps or flood risk information, and how to make it more easily accessible. Finally, whereas in the past it was not feasible to present the public with detailed, property-level flood risk data, modern technology and data intensive applications now allow for this functionality, and this is an opportunity for governments to provide meaningful information to support the public in disaster risk reduction.

**Acknowledgements**

To be added.





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
