# Peer review of "Communicating Disaster Risk? An Evaluation of the Availability and Quality of Flood Maps"

_Natural Hazards and Earth System Sciences, 2018_

## Referee Comment (RC1) · M. Nones (Referee) · 17 Oct 2018

Using Canada as a case study, the authors evaluated the availability and the quality of flood maps, in a first attempt to classify them and suggest possible improvements for better communicating the risk to stakeholders and citizens. As pointed out by other studies worldwide, there are several issues related to flood maps, spanning from their design to their accessibility, and therefore any research in this sense is meaningful, and shall definitely be published in NHESS.

The literature review is very good, especially as Canada is concerned, and I have only a few hints to improve it. Firstly, additional insights on the use of flood hazard maps in the US and in Europe are discussed, as an example, by Luke et al. (2018), Albano et

al. (2017) and Nones (2017), pointing out a generalized lack of consistency and the urgency in moving towards a new approach in communicating flood risk. Moreover, as for improving the discussion about the impact of floods on critical infrastructures and how to communicate risks not easily catchable by citizens, I would like to suggest the recent work made by Serre and Heinzlef (2018) on urban environments.

As for the methods, I clearly understand why the authors focussed on only 369 FDPR communities, but I am wondering if studying 1/3 of all the communities could lead to some biases. I do not see any discussion on this assumption along the manuscript, therefore I recommend adding some comments. In this context, there is a project to extend the analysis to the whole of Canada? Could be a huge work, but definitely worth of meaning for addressing the challenge of risk communication.

I can imagine that the searching for the maps, their comparison and their evaluation lasted several months. You said that the search was "concluded on July, 25 2018" [page 8, line 15], but you are not saying when it started. In other words, could be the time an important factor in such studies? Are you sure that "inaccessible" maps at the beginning of the search were still inaccessible in July? Probably yes, but a discussion in this sense can be helpful

Under a general point of view, the results reported here are very interesting and in line with other studies, showing how challenging the topic is. I encourage the authors to further develop the research, given that has the potential to become fundamental in addressing the topic of risk flood communication.

A few additional minor comments and technical corrections:

Table 1 and Table 2 can be combined, showing the percentage of each FDPR communities analysed in each territory.

Table 3: change the caption to something like "map assessment criteria and sources".

As for Figure 3, stay with the percentage of municipalities instead of the number, to be

consistent with Figure 1.

References

Albano R., Mancusi L., Abbate A. (2017). Improving flood risk analysis for effectively supporting the implementation of flood risk management plans: The case study of "Serio" Valley. Environmental Science & Policy 75, 158-172. doi: 10.1016/j.envsci.2017.05.017

Luke A., Sanders B.F., Goodrich K.A., Feldman D.L., Boudreau D., Eguiarte A., Serrano K., Reyes A., Schubert J.E., AghaKouchak A., Basolo V., Matthew R.A. (2018). Going beyond the flood insurance rate map: insights from flood hazard map co-production. Nat. Hazards Earth Syst. Sci., 18, 1097-1120. doi: 10.5194/nhess-18-1097-2018

Nones M. (2017). Flood hazard maps in the European context. Water International 42(3), 324-332. doi: 10.1080/02508060.2016.1269282

Serre D., Heinzlef C. (2018). Assessing and mapping urban resilience to floods with respect to cascading effects through critical infrastructure networks. Int. Journal of Disaster Risk Reduction 30(B), 235-243. doi: 10.1016/j.ijdrr.2018.02.018

---

## Author Comment (AC1) · 23 Oct 2018

We are pleased to respond to the helpful and constructive comments of Referee 1 (Nones), which were posted on the NHESS Discussion page on October 17, 2018. The Referee's comments and our responses are presented below.

1. The literature review is very good, especially as Canada is concerned, and I have only a few hints to improve it. Firstly, additional insights on the use of flood hazard maps in the US and in Europe are discussed, as an example, by Luke et al. (2018), Albano et al. (2017) and Nones (2017), pointing out a generalized lack of consistency and the urgency in moving towards a new approach in communicating flood risk. Moreover, as for improving the discussion about the impact of floods on critical infrastructures and

[Figure]

how to communicate risks not easily catchable by citizens, I would like to suggest the recent work made by Serre and Heinzlef (2018) on urban environments.

RESPONSE: We are grateful to Referee 1 for pointing out these additional sources, which appear highly relevant to our study and will enrich its resource base. We will integrate these papers into the revised manuscript submitted at the end of the Discussion stage.

2. As for the methods, I clearly understand why the authors focussed on only 369 FDRP communities, but I am wondering if studying 1/3 of all the communities could lead to some biases. I do not see any discussion on this assumption along the manuscript, therefore I recommend adding some comments. In this context, there is a project to extend the analysis to the whole of Canada? Could be a huge work, but definitely worth of meaning for addressing the challenge of risk communication.

RESPONSE: Our original ambition for this study was to systematically assess the availability and quality of flood maps across all 957 Canadian communities labelled as "designated (flood risk) areas" under the Flood Damage Reduction program. However, upon discovering the labour required to locate and code the maps, we decided instead to code a sample. Using a 95% confidence interval and 4% margin of error, we drew a random sample of 369 communities. Among this sample, the percentage of maps from each individual province roughly approximated the percentage of maps per province in the total dataset. As such, we are confident that the results generated from coding this sample of maps are generalizable to all 957 FDRP-designated communities. We agree with Referee 1 that a Canada-wide analysis would provide a more fulsome picture. Although it is not feasible for this study, we hope to extend the analysis in future. We will clarify these points in the revised manuscript.

3. I can imagine that the searching for the maps, their comparison and their evaluation lasted several months. You said that the search was "concluded on July, 25 2018" [page 8, line 15], but you are not saying when it started. In other words, could be the

time an important factor in such studies? Are you sure that "inaccessible" maps at the beginning of the search were still inaccessible in July? Probably yes, but a discussion in this sense can be helpful.

RESPONSE: We agree with Referee 1 that it is useful to provide more clarity on the methods here. Once the random sample of maps was drawn, we searched until we were confident that we had collected all accessible maps, and this process lasted for 6.5 weeks (June 18 to July 25). We were unable to find maps in 41 municipalities (15%) in the sample set. We are confident that these communities did not produce and publish maps within the 6.5-week period (and were therefore miscoded as "inaccessible"), because to our knowledge none of the four provinces in which these communities are located (Saskatchewan, Nova Scotia, Ontario and Manitoba) had active mapping efforts underway at the time. We will add this explanation to the revised manuscript before resubmission.

4. Under a general point of view, the results reported here are very interesting and in line with other studies, showing how challenging the topic is. I encourage the authors to further develop the research, given that has the potential to become fundamental in addressing the topic of risk flood communication.

RESPONSE: We appreciate Referee 1's encouraging comments on the utility of this study, and we hope to extend this analysis in future research.

5. A few additional minor comments and technical corrections:

(a) Table 1 and Table 2 can be combined, showing the percentage of each FDPR communities analysed in each territory.

RESPONSE: We agree with Referee 1 that Tables 1 and 2 could be combined so that the reader can compare the number and percentage of maps per province in the total set vs. the number and percentage of maps per province in the sample set. We will make this change in the revised manuscript before resubmission.

(b) Table 3: change the caption to something like "map assessment criteria and sources".

RESPONSE: We will adjust the title of Table 3 to acknowledge that it also contains the sources of the various evaluation criteria. We will make this change in the revised manuscript before resubmission.

(c) As for Figure 3, stay with the percentage of municipalities instead of the number, to be consistent with Figure 1.

RESPONSE: We agree with Referee 1 that Figure 3 should be changed to reflect the percentage of municipalities (rather than the number of municipalities) that meet the various evaluation criteria, in order to be consistent with Figure 1. We will make this change in the revised manuscript before resubmission.

---

## Referee Comment (RC2) · S. Fuchs (Referee) · 15 Nov 2018

Referee report on "Communicating Disaster Risk? An Evaluation of the Availability and Quality of Flood Maps" by Henstra et al.

The authors assess the availability and quality of flood hazard and flood risk maps for communities in Canada, focusing explicitly on the communication to the general public. As such, the topic is in the scope of the target journal and of valuable importance also from a scientific point of view. The article is well-written, based on up-to-date research methods and through its structure also very accessible.

I only have some minor comments that may enrich the content in one or the other way:

- Page 3, lines 25 ff.: Quoting the paper of de Moel et al. the authors state that 29

[Figure]

European countries already have flood maps but only very few have produced flood risk maps that include information on the consequences of flooding. It may be worth to add here some sentences on the European Flood Directive and its implementation: Increasing flood losses throughout Europe have led the European Commission to issue the Directive on the Assessment and Management of Flood Risks (Commission of the European Communities, 2007) as one of the three components of the European Action Program on Flood Risk Management (Commission of the European Communities, 2004). This directive, defining flood hazard in the broadest terms as "the temporary covering by water of land not normally covered by water" requires the member states to establish flood risk maps and flood risk management plans based on a nationwide evaluation of hazard, exposure, and vulnerability (e.g., Fuchs et al., 2017). While in the early 21st century considerable efforts have been made toward flood risk maps (Meyer et al., 2012), less information is available so far on respective management plans (Hartmann and Spit, 2016). Moreover, there is a particular gap in risk maps and management plans for mountain hazards other than those of hydrological origin. Of particular importance seems to be the paradigm of public participation and societal adaptation in assessing local risks, and the legal and institutional settings necessary therefore (Hartmann and Driessen, 2017; Thaler et al., 2018).

- Figure 2: Please think about enlarging the Figure so that the readers can follow your arguments regarding "bad practice" and "good practice" – alternatively, you may wish to insert one "best practice" example in section 5.2.

- Page 14, lines 24 ff.:The mentioned shift towards more self-responsibility in mitigation and adaptation decisions is also because of a decreasing budget available for technical mitigation – you may wish to check (again with a focus on the European Alps) Holub and Fuchs (2009) how these issues can be put together so that the overall societal resilience is increased. Moreover, to show property-level flood risk publicly has been heavily debated in Europe because of protection of data privacy. As such, some European websites restrict the zoom function to a certain scale so that not everybody

can precisely assess the hazard extent and match this information with the real estate extent (for an example of limited zoom possibilities, see https://www.hora.gv.at/).

References mentioned (for illustration purpose only, please note that it is up to the authors whether or not they wish to include them into their revised manuscript)

Commission of the European Communities: Communication from the Commission to the Council, the European Parliament, the European Economic and Social Committee and the Committee of the Regions - Flood risk management - Flood prevention, protection and mitigation, COM(2004) 472 final of 12.7.2004, 2004.

Commission of the European Communities: Directive 2007/60/EC of the European Parliament and of the Council of 23 October 2007 on the assessment and management of flood risks, Official Journal of the European Union, L 288, 27-34, 2007.

Fuchs, S., Röthlisberger, V., Thaler, T., Zischg, A., and Keiler, M.: Natural hazard management from a coevolutionary perspective: Exposure and policy response in the European Alps, Annals of the American Association of Geographers, 107, 382-392, 2017.

Hartmann, T., and Driessen, P. P.: The flood risk management plan: towards spatial water governance, Journal of Flood Risk Management, 10, 145-154, 2017.

Hartmann, T., and Spit, T.: Implementing the European flood risk management plan, Journal of Environmental Planning and Management, 59, 360-377, 2016.

Holub, M., and Fuchs, S.: Mitigating mountain hazards in Austria – Legislation, risk transfer, and awareness building, Natural Hazards and Earth System Sciences, 9, 523-537, 2009.

Meyer, V., Kuhlicke, C., Luther, J., Fuchs, S., Priest, S., Dorner, W., Serrhini, K., Pardoe, J., McCarthy, S., Seidel, J., Scheuer, S., Palka, G., Unnerstall, H., and Viavatenne, C.: Recommendations for the user-specific enhancement of flood maps, Natural Hazards and Earth System Sciences, 12, 1701-1716, 2012.

Interactive
comment

[Figure]

Thaler, T., Zischg, A., Keiler, M., and Fuchs, S.: Allocation of risk and benefits – distributional justices in mountain hazard management, Regional Environmental Change, 18, 353-365, 2018.

---

## Author Comment (AC2) · 22 Nov 2018

We are pleased to respond to the helpful and constructive comments of Referee 2 (Fuchs), which were posted on the NHESS Discussion page on November 15, 2018. The Referee's comments and our responses are presented below.

1. Page 3, lines 25 ff.: Quoting the paper of de Moel et al. the authors state that 29 European countries already have flood maps but only very few have produced flood risk maps that include information on the consequences of flooding. It may be worth to add here some sentences on the European Flood Directive and its implementation: Increasing flood losses throughout Europe have led the European Commission to issue the Directive on the Assessment and Management of Flood Risks (Commission of the

European Communities, 2007) as one of the three components of the European Action Program on Flood Risk Management (Commission of the European Communities, 2004). This directive, defining flood hazard in the broadest terms as "the temporary covering by water of land not normally covered by water" requires the member states to establish flood risk maps and flood risk management plans based on a nationwide evaluation of hazard, exposure, and vulnerability (e.g., Fuchs et al., 2017). While in the early 21st century considerable efforts have been made toward flood risk maps (Meyer et al., 2012), less information is available so far on respective management plans (Hartmann and Spit, 2016). Moreover, there is a particular gap in risk maps and management plans for mountain hazards other than those of hydrological origin. Of particular importance seems to be the paradigm of public participation and societal adaptation in assessing local risks, and the legal and institutional settings necessary therefore (Hartmann and Driessen, 2017; Thaler et al., 2018).

RESPONSE: We agree with Referee 2 that it is important to highlight the EU Floods Directive as a key catalyst for flood risk mapping and flood risk management plans in Europe. The revised manuscript—to be submitted after the Discussion Period—will contain a few sentences to explain the Directive and, drawing on the sources recommended by Referee 2, provide a more recent assessment of progress toward its implementation, including outstanding challenges.

2. Figure 2: Please think about enlarging the Figure so that the readers can follow your arguments regarding "bad practice" and "good practice" – alternatively, you may wish to insert one "best practice" example in section 5.2.

RESPONSE: In the revised manuscript, we will expand Figure 2 to better illustrate the differences in quality between the various maps. We will also include an example of a "best practice" map—one scoring 78% (7/9), which was the highest assigned score among the dataset.

3. Page 14, lines 24 ff.: The mentioned shift towards more self-responsibility in mitigation and adaptation decisions is also because of a decreasing budget available for technical mitigation – you may wish to check (again with a focus on the European Alps) Holub and Fuchs (2009) how these issues can be put together so that the overall societal resilience is increased.

RESPONSE: In the revised manuscript, we will note that risk transfer policies that shift responsibility to individuals are often motivated by declining budgets for structural protections, and that increasing risk awareness through information is an essential prerequisite for their success.

4. Moreover, to show property-level flood risk publicly has been heavily debated in Europe because of protection of data privacy. As such, some European websites restrict the zoom function to a certain scale so that not everybody can precisely assess the hazard extent and match this information with the real estate extent (for an example of limited zoom possibilities, see https://www.hora.gv.at/).

RESPONSE: This is an important point, and we are grateful to Referee 2 for raising it. Whether and how releasing flood maps could affect property values or data privacy is a real concern, and one that has not been addressed fully in the Canadian context. Although this discussion is outside the scope of our current paper, we will make note of this issue as we continue with this project based on feedback from policy-makers and practitioners.

5. References mentioned (for illustration purpose only, please note that it is up to the authors whether or not they wish to include them into their revised manuscript)

Commission of the European Communities: Communication from the Commission to the Council, the European Parliament, the European Economic and Social Committee and the Committee of the Regions - Flood risk management - Flood prevention, protection and mitigation, COM(2004) 472 final of 12.7.2004, 2004.

Commission of the European Communities: Directive 2007/60/EC of the European

Parliament and of the Council of 23 October 2007 on the assessment and management of flood risks, Official Journal of the European Union, L 288, 27-34, 2007.

Fuchs, S., Röthlisberger, V., Thaler, T., Zischg, A., and Keiler, M.: Natural hazard management from a coevolutionary perspective: Exposure and policy response in the European Alps, Annals of the American Association of Geographers, 107, 382-392, 2017.

Hartmann, T., and Driessen, P. P.: The flood risk management plan: towards spatial water governance, Journal of Flood Risk Management, 10, 145-154, 2017.

Hartmann, T., and Spit, T.: Implementing the European flood risk management plan, Journal of Environmental Planning and Management, 59, 360-377, 2016.

Holub, M., and Fuchs, S.: Mitigating mountain hazards in Austria – Legislation, risk transfer, and awareness building, Natural Hazards and Earth System Sciences, 9, 523-537, 2009.

Meyer, V., Kuhlicke, C., Luther, J., Fuchs, S., Priest, S., Dorner,W., Serrhini, K., Pardoe, J., McCarthy, S., Seidel, J., Scheuer, S., Palka, G., Unnerstall, H., and Viavatenne, C.: Recommendations for the user-specific enhancement of flood maps, Natural Hazards and Earth System Sciences, 12, 1701-1716, 2012.

Thaler, T., Zischg, A., Keiler, M., and Fuchs, S.: Allocation of risk and benefits – distributional justices in mountain hazard management, Regional Environmental Change, 18, 353-365, 2018.

RESPONSE: We are grateful to Referee 2 for suggesting these resources, most of which we have incorporated into the revised paper.